# *Maytenus octogona* Superoxide Scavenging and Anti-Inflammatory Caspase-1 Inhibition Study Using Cyclic Voltammetry and Computational Docking Techniques

**DOI:** 10.3390/ijms241310750

**Published:** 2023-06-28

**Authors:** Francesco Caruso, Miriam Rossi, Eric Eberhardt, Molly Berinato, Raiyan Sakib, Felipe Surco-Laos, Haydee Chavez

**Affiliations:** 1Department of Chemistry, Vassar College, Poughkeepsie, NY 12604, USA; 2Facultad de Farmacia y Bioquímica, Universidad Nacional San Luis Gonzaga, Ica 11004, Peru

**Keywords:** *Maytenus*, caspase-1, inflammation, antioxidants, tingenone, pristimerin, hydroxytingenone, docking, DFT, hydrodynamic voltammetry

## Abstract

The relationship between oxidative stress and inflammation is well known, and exogenous antioxidants, primarily phytochemical natural products, may assist the body’s endogenous defense systems in preventing diseases due to excessive inflammation. In this study, we evaluated the antioxidant properties of ethnomedicines from Peru that exhibit anti-inflammatory activity by measuring the superoxide scavenging activity of ethanol extracts of *Maytenus octogona* aerial parts using hydrodynamic voltammetry at a rotating ring-disk electrode (RRDE). The chemical compositions of these extracts are known and the interactions of three methide-quinone compounds found in *Maytenus octogona* with caspase-1 were analyzed using computational docking studies. Caspase-1 is a critical enzyme triggered during the activation of the inflammasome and its actions are associated with excessive release of cytokines. The most important amino acid involved in active site caspase-1 inhibition is Arg341 and, through docking calculations, we see that this amino acid is stabilized by interactions with the three potential methide-quinone *Maytenus octogona* inhibitors, hydroxytingenone, tingenone, and pristimerin. These findings were also confirmed after more rigorous molecular dynamics calculations. It is worth noting that, in these three compounds, the methide-quinone carbonyl oxygen is the preferred hydrogen bond acceptor site, although tingenone’s other carbonyl group also shows a similar binding energy preference. The results of these calculations and cyclovoltammetry studies support the effectiveness and use of anti-inflammatory ethnopharmacological ethanol extract of *Maytenus octogona* (L’Héritier) DC.

## 1. Introduction

Caspases are a highly conserved family of cysteine protease enzymes that may induce apoptosis and inflammation [1]. Caspase-1 plays a crucial role in the inflammatory response by cleaving proteins after triggering the release of proinflammatory cytokines such as interleukin-1 beta (IL-1β) and interleukin-18 (IL-18). During a viral attack, the immune system reacts strongly to this invasion, sometimes overproducing cytokines—a “cytokine storm”. Events associated with the over-activation of the immune response have been connected to poor outcomes in patients with COVID-19 [2,3]. For instance, high levels of proinflammatory cytokine IL-1β were detected in the lung parenchyma of COVID-19 patients [4]. Moreover, increased levels of IL-18 are observed with increased mortality in sepsis-induced acute respiratory distress syndrome (ARDS) [5]. Inhibition of caspase-1 and the inflammasome pathway may provide an opportunity to suppress the heightened immune response and offer an alternative anti-COVID-19 therapy [6].

Activation of the inflammasome is a crucial function mediated by the innate immune system [3], Appendix A. Initiation is necessary in cases of cellular or molecular damage or, on infection, when interactions with foreign pathogens occur. Since inflammation is associated with many human diseases, anomalous activation can result in uncontrolled responses responsible for many pathogenic states besides the recent COVID-19 pandemic described earlier. The initial step of inflammasome activation involves the binding of exogenous ligands, such as pathogen-associated molecular patterns (PAMPs) or damage/danger-associated molecular patterns (DAMPs), to a pattern recognition receptor (PRR). Several families of PRRs exist, including the nucleotide-binding domain and leucine-rich repeat-containing proteins (NLRs, also known as NOD-like receptors). PRR binding stimulates a cascade that produces pro-IL-1β and pro-IL-18 in an NFKβ-dependent manner [7], with the relevant NLR oligomerizing becoming a caspase-1-activating scaffold. Active caspase-1 amplifies the inflammasome signal by cleaving the proinflammatory IL-1 family of cytokines into their bioactive forms, IL-1β and IL-18, and causes pyroptosis, a type of inflammatory cell death [8,9]. A second signal, from various origins, involves releasing reactive oxygen species (ROS) that stimulate inflammasome assembly, leading to the activation of caspase-1 and the subsequent processing of pro-IL-1β into IL-1β and pyroptosis [3]. In some cases, with excessive amounts of cytokines being produced, diseases such as atherosclerosis, cancer, and COVID-19 infection can result. Of late, several known anti-inflammatory medicinal compounds are being reevaluated to counteract the body’s inflammatory response to COVID-19 virus infection. Antioxidants that suppress ROS production and caspase-1 activation may help diminish the inflammasome response and are promising for the development of a novel therapy. Details of caspase-1 role in inflammation have been described [6], including a recent review [10].

Recently, to analyze the inhibition of caspase-1 by anti-inflammatory drugs, our group used computational methods and measured the antioxidant activity towards the damaging superoxide radical using the RRDE cyclovoltammetry method that was developed in our laboratory [11]. Previously, we associated the anti-COVID-19 biological activity of celastrol, a natural product abundant in flowering plants of the genus *Maytenus* and the family *Celastraceae*, with its inhibition of the COVID-19 main protease and its scavenging of superoxide [12].

The roots, bark, and leaves of *Maytenus* plants, spread in tropical and sub-tropical regions of the world, are a rich source of widely used triterpenoids in ethnopharmacology and traditional medicines as anti-tumor, anti-asthmatic, analgesic, anti-inflammatory, antimicrobial, and antiulcer agents [13,14]. These plants are common in Peru, a country that has a rich flora that provides many useful bioactive compounds [15]. Indigenous populations use these plants for their anti-inflammatory properties. In particular, the high sub Andean region of the Amazonian river basin (Peru, Ecuador, Colombia) is home to a *Maytenus* species named chuchuasha (or chuchuaso), whose alcoholic root bark extracts are used in traditional medicine to reduce inflammation. Because a relationship between oxidative stress and inflammation is known, exogenous antioxidants may provide a potential protective effect by assisting the body’s defense systems in preventing diseases due to excessive inflammation. Phenoldienones such as tingenone and 2-hydroxytingenone are present in the plant, and the biological activities of these compounds are in agreement with those observed when using the plant extracts in traditional health practices [16].

Following our interest in investigating the medicinal properties of the *Maytenus* genus [17,18,19,20], in this study, we analyzed the antioxidant properties of ethanol extracts from the leaves and stems of *Maytenus octogona* plants. We also focused on three components related to celastrol from these plants—tingenone, hydroxytingenone, and pristimerin—and studied their interactions with caspase-1 by performing docking studies to analyze their potential inhibitory effects. Figure 1 shows the studied molecules. The scavenging of superoxide by *Maytenus octogona* ethanol extracts was also described in this study. The samples used in this study were collected by biologist Alfonso Orellana García in the area of Molletambo, district of Yauca del Rosario, Region and Department of Ica, at georeferencing points (GPS): latitude (S): 14°11′1.73″, longitude (W): 75°22′47.28″, Voucher AO-113 (UNICA).

## 2. Results and Discussion

### 2.1. Caspase-1 Docking

#### 2.1.1. Hydroxytingenone

Using Materials Studio Dmol^3^ quantum chemistry program, the molecular structure of hydroxytingenone was obtained through modifying the related crystal structure of celastrol [12]. Molecular geometry optimization of the structure did not include the solvent effect. This was suggested by the absence of water molecules in the active site and the resulting hydroxytingenone coordinates were input for the docking study, which was performed using the Discovery Studio molecular mechanics program on caspase-1. From the Protein Data Bank (PDB) [21], we retrieved the PDB 6PZP caspase-1 in complex with a small molecule inhibitor, VX-765 [22], where caspase-1 consisted of two protein subunits. After Discovery Studio application of the CHARMm forcefield [23], H atoms were assigned, VX-765 was selected to create a sphere of radius 10 Å and was then removed. Hence, hydroxytingenone was put in its place, docked, and 10 poses were obtained.

The relevant moieties of hydroxytingenone able to interact with the active site of caspase-1 are the two carbonyl groups, located at both ends of the structure, as discussed previously [11]. The carbonyl at the methide-quinone end is found in pose 4. A two-dimensional display of the amino acid interactions of docked hydroxytingenone at the active site is shown in Figure 2. Discovery Studio provides the Cdocker interaction energy, which is the non-bonded interaction energy (composed of the van der Waals term and the electrostatic term) between the protein and the potential inhibitor ligand related to the force field CHARMm; this value is −22.4 kcal/mol for hydroxytingenone pose 4. A more specific display of pose 4 is shown in Figure 3, where besides the H-bonds relevant to the methide-quinone moiety (Arg341 and His237), a Cys285 interaction (upon a slight variation in the cysteine C-C-S-H torsion angle) is seen to establish a very short H-bond distance between H(Cys285) and the hydroxytingenone hydroxyl group, 1.837 Å. However, the Cys285 interaction is not confirmed after a standard dynamic cascade protocol of Discovery Studio, while Arg341 and His237 are indeed confirmed. The final arrangement, including calculating binding energy (−13.7 kcal/mol), is shown in Figure 4.

Another cluster that includes poses 5, 6, 7, 9 shows important interactions at the opposite end of hydroxytingenone, with Cys285 forming a H-bond to the O(carbonyl). Pose 5 was not confirmed by standard dynamics cascade as the ligand was thoroughly displaced from the active site. A smaller cluster was formed by poses 1 and 2, where Cys285 and His237 show H-bonds to the carbonyl opposite the methide-quinone moiety. After applying dynamics to pose 1, the binding energy is −7.2 kcal/mol, which is markedly more positive than that of pose 4 (−13.7 kcal/mol). Additionally, pose 3 shows an interaction between the same carbonyl involved in pose 1, having H-bonds with Cys285 and Arg341. After dynamics, the binding energy is −8.9 kcal/mol, slightly better than that of pose 1 but still much less favored than that of pose 4. We conclude that hydroxytingenone molecular mechanics favor a pose 4 arrangement, shown in Figure 4.

#### 2.1.2. Tingenone

Our next docked compound was tingenone. Its atomic coordinates were derived from hydroxytingenone and minimized with Dmol^3^. Poses 1 and 5 are representatives of well-distinguished clusters in the docking as the first includes poses 2–4, 6, and 8, whereas pose 5 cluster includes poses 7 and 9. Pose 1 has Cdocker interaction energy −27.9 kcal/mol and shows interactions with the non-methide-quinone carbonyl, involving Arg341 and His237, see Figure 5 and Appendix A. After the standard dynamics cascade, pose 1 is confirmed and the final arrangement obtained after binding energy calculations, −11.8 kcal/mol, is shown in Figure 6; this does not include His237. Interestingly, a cation-π interaction from another arginine, Arg383, is present at the other end of the molecule involving the methide-quinone moiety (2.752 Å) and so this pose appears tightly bound at the active site.

The only H-bond shown by pose 5 docking (Cdocker interaction energy = −25.1 kcal/mol) is with Arg341, which also involves the non-methide-quinone moiety (Figure 7 and Appendix A), which is confirmed by dynamics and has a calculated binding energy of −13.0 kcal/mol, Figure 8. Thus, both carbonyls of tingenone show a similar preference due to the equal binding energy of poses 1 and 5.

#### 2.1.3. Pristimerin

The last docked compound was pristimerin. Pose 4 belongs to the most populated cluster that includes poses 7 and 10. Its Cdocker interaction energy is −26.8 kcal/mol. There is a methide-quinone moiety involved in this cluster, interacting with Arg341 and π-sulfur Cys285 Figure 9 and Appendix A. Dynamics confirms the arrangement and, after calculating binding energy, −9.4 kcal/mol, Figure 10 shows three successive amino acid residues involved, Ser339-Trp340-Arg341, forming H-bonds of 1.524 Å, 2.297 Å and 2.370 Å, respectively.

The carbonyl (non methide-quinone) located at the other end of pristimerin (pose 1) shows H-bond interactions with Cys285 and Arg341, which is not confirmed by applying dynamics as pristimerin moves out of the active site. When analyzing the cluster made by poses 2 and 3, an H-bond interaction with Asp288 is seen. Upon dynamics of pose 2, this evolves towards an H-bond with Cys285, whose binding energy is −9.3 kcal/mol, similar to that arising from pose 4. No other poses involve this carbonyl opposite the methide-quinone.

Poses 5 and 8 show π-π interactions with His342 and the dynamics of pose 5 displays no H-bonds; the binding energy of this conformation is −5.8 kcal/mol. Additionally, pose 6 was analyzed and showed only van der Waals interactions. Upon dynamics, Gly238 shows an H-bond and a binding energy of −5.8 kcal/mol. Therefore, pristimerin pose 4, involving the methide-quinone carbonyl, and pose 2, interacting with the opposite carbonyl, are equally preferred: −9.4 and −9.3 kcal/mol, respectively. A summary of amino acid interactions is displayed in Table 1.

From this docking study on these three natural products, we see marked differences to our earlier published findings on dexamethasone and other anti-inflammatory drugs [11] that showed important interactions with Cys285. In the present study, Cys285 shows no interaction with tingenone at docking, whereas hydroxytingenone and pristimerin do show such an interaction, although these are not confirmed after more rigorous molecular dynamics calculations. Here, Arg341 is the more involved amino acid at the caspase-1 active site as it is stabilized by the three potential inhibitors, hydroxytingenone, tingenone, and pristimerin. This interaction is also present after dynamics. In the three natural products, we see the methide-quinone carbonyl group to be the preferred acceptor involved in hydroxytingenone, tingenone, and pristimerin hydrogen bonding interactions, agreeing with our previous findings [11]; however, the other carbonyl in tingenone shows a similar preference. Moreover, the interactions of our three celastrol derivatives docked at the active site of caspase-1 involve amino acids also engaged by the inhibitor VX-765 in the crystal structure, suggesting a closely related mechanism of inhibition. This includes hydrogen bonds for all three analyzed ligands with Arg341; cation-π interaction with Arg383, established with an aromatic ring of tingenone, and hydrogen bond interaction with O(Ser339) for hydroxytingenone and pristimerin.

### 2.2. RRDE

The release of ROS, a signal for inflammasome activation, may come from a variety of pathways, including K^+^ efflux, lysosomal rupture or mitochondrial dysfunction [6]. Thus, analyzing the antioxidant capability of the plant extracts may provide an explanation for their medicinal use.

In our laboratory, we developed an electrochemical method for the direct and quantitative analysis of scavenging superoxide [24]. Figure 11 shows a collage of the voltammograms associated with such an experiment for sample MORUnica (stem extract). The bottom part of the figure shows the amount of superoxide generated after reduction of bubbled O_2_ in the electrochemical cell (reaction (1), see Section 3.3); this is detected at the disk electrode. The top part of the figure shows superoxide detected at the ring electrode. The first experiment is a blank indicated as a 0 µL line; the last one shows the minimum amount of detected superoxide, for aliquot 510 µL. Each added aliquot decreases the current intensity detected at the ring electrode, and this is due to superoxide consumed by the antioxidant components of the extracts. For each voltammogram, the ratio ring current/disk current is defined as the efficiency. The physical interpretation is also associated with the time the generated superoxide (at the disk) takes to reach the ring and be oxidized.

In Figure 12, the efficiency of all added aliquots is included, i.e., the collection efficiency is shown at the y axis, while the x axis shows the volume of added aliquot; ultimately, the slope of the line is associated with the antioxidant capability of scavenging. Figure 13 and Figure 14 show equivalent graphs for MOHUnica (leaf extract). Table 2 includes slopes of other natural products studied using the same method. This shows both *Maytenus octogona* extracts having weaker antioxidant activity, compared with olive oil. This is probably due to the depletion of antioxidants during ethanolic sample preparations of *Maytenus octogona* extracts, in contrast with cold-extracted extra virgin olive oil [25].

## 3. Materials and Methods

### 3.1. Reagents

For electrochemical studies, tetrabutylammonium bromide (TBAB; TCI Chemicals, Portland, OR, USA) and 99.9% anhydrous dimethyl sulfoxide (DMSO; Sigma-Aldrich, Inc., St. Louis, MO, USA).

#### 3.1.1. MOH UNICA (Leaf Extract)

Two methods were used to obtain these samples:

Method 1. Two kilograms of dry and ground *Maytenus octogona* leaves were macerated for 20 h in hot (96°) ethanol and were filtered. After this time, additional ethanol was added and the mixture was refluxed for 4 h. The solution was filtered hot.

Method 2. One kilogram of dried and ground leaves of *Maytenus octagona* were extracted from ethanol 96% in a Soxhlet apparatus until exhaustion. Evaporation of the solvent at reduced pressure provided 260 g of a dark green extract.

Both the filtrates were combined and concentrated in a Buchi rotary evaporator at 40 °C and 400 g of a dark green dry extract was obtained.

#### 3.1.2. MOR UNICA (Stem Extract)

One kilogram of dried and ground stems of *Maytenus octagona* was extracted from ethanol 96% in a Soxhlet apparatus until exhaustion. Evaporation of the solvent at reduced pressure provided 275 g of a reddish-brown extract.

### 3.2. Electrochemistry

A Pine Research WaveDriver 20 bipotentiostat with the Modulated Speed Electrode Rotator was used to perform the hydrodynamic voltammetry at a rotating ring-disk electrode (RRDE). The working electrode is the AFE6R2 gold disk and gold ring rotator tip (Pine Research, Durham, NC, USA) combined with a coiled platinum wire counter electrode and a reference electrode consisting of an AgCl coated silver wire immersed in 0.1 M tetrabutyl ammonium bromide (TBAB) in dry DMSO in a fritted glass tube. The electrodes were placed in a five-neck electrochemical cell together with means for either bubbling or blanketing the solution with gas. Voltammograms were collected using Aftermath software release 1.6.10523 provided by Pine Research. Careful cleaning of the electrodes was performed by polishing with 0.05 µm alumina-particle suspension (Allied High Tech Products, Inc., Rancho Dominguez, CA, USA) on a moistened polishing microcloth to eliminate potential film formation [28].

### 3.3. Hydrodynamic Voltammetry (RRDE)

MOH Unica (0.924 g) and MOR Unica (0.103 g) were dissolved in 8.7 mL and 9.0 mL, respectively, of anhydrous DMSO (99.9% purity). These were the stock solution of analyzed samples. For the experiment, a solution of 0.1 M TBAB electrolyte in anhydrous DMSO (99.9% purity) was bubbled for 5 min with a dry O_2_/N_2_ (35%/65%) gas mixture to establish the dissolved oxygen level in the electrochemical cell of 50 mL. The Au/Au disc electrode was then rotated at 1000 rpm, and potential sweep was applied to the disk from 0.2 V to −1.2 V and then back to 0.2 V while the ring was held constant at 0.0 V; the disk voltage sweep rate was set to 25 mV/s. The molecular oxygen reduction peak (reaction 1) was observed around −0.6 V at the disk electrode; the oxidation current (reaction 2) occurred at the ring electrode. An initial blank, in the absence of an antioxidant, was run on this solution and the ratio of the ring/disk current was calculated as the “efficiency”. Next, an antioxidant aliquot was added, the solution bubbled with the gas mixture for 5 min, the voltammogram was rerecorded, and was efficiently obtained. In this way, the rate at which increasing concentrations of antioxidant scavenge the generated superoxide radicals during the electrochemical reaction was determined as each additional antioxidant aliquot was added. Results from each run were collected on Aftermath software Release 1.6.10523 and represented as voltammograms showing current vs. potential graphs that were later analyzed using Microsoft Excel. The used aliquots were indicated in related RRDE graphs. Ultimately, the slope of the overall decrease in efficiency with the addition of antioxidant serves as a quantitative measure of the antioxidant activity of each sample. Any decrease in the collection efficiency was expected to be due to the amount of superoxide removed by the antioxidant. This method has been developed in our lab [24].

In an RRDE voltammetry experiment, the generation of the superoxide radicals occurs at the disk electrode while the oxidation of the residual superoxide radicals (that have not been scavenged by the antioxidant) occurs at the ring electrode.

Reaction (1): Reduction of molecular oxygen at the disk electrode:O_2_ + e^−^ → O_2_•^−^(1)

Reverse Reaction (2): Oxidation of superoxide radicals at the ring electrode:O_2_•^−^ → O_2_ + e^−^(2)

### 3.4. Computational Study

Calculations were performed using programs from Biovia (Dassault Systèmes, San Diego, CA, USA). Density functional theory (DFT) program Dmol3 was applied to calculate energy, geometry, and frequencies implemented in Materials Studio 7.0 [29]. We employed the double numerical polarized (DNP) basis set that included all the occupied atomic orbitals plus a second set of valence atomic orbitals and polarized d-valence orbitals [30]; the correlation generalized gradient approximation (GGA) was applied, including Becke exchange [31] plus BLYP correlation. All electrons were treated explicitly and the real space cutoff of 5 Å was imposed for numerical integration of the Hamiltonian matrix elements. The self-consistent field convergence criterion was set to the root mean square change in the electronic density to be less than 10^−6^ electron/Å^3^. Calculations did not include solvent effects. The convergence criteria applied during geometry optimization were 2.72 × 10^−4^ eV for energy and 0.054 eV/Å for force. Docking studies were performed with the CDOCKER package in Discovery Studio 2020 version [23]. Standard dynamics cascade protocol in Discovery Studio allows optimization of atomic coordinates and was also applied to the selected poses.

## 4. Conclusions

Since inflammation is associated with many human maladies including metabolic and cardiovascular diseases, cancer, and COVID-19, as well as many neurodegenerative conditions, inhibitors of the anomalous activation of the inflammasome are important to identify [32]. Selective, potent small molecule caspase-1 inhibitors such as VX-765 (Belnacasan) exist but their strong side effects limit their clinical use [33]. Recent in vivo and in vitro studies on a small molecule caspase-1 inhibitor, NSC697923, showed its effectiveness in an animal model of gouty arthritis. When docking studies were performed that included NSC697923 in the binding pocket A of caspase-1, H-bonds with Arg341 were seen and, consistent with VX-765, the caspase-1 inhibitor was present in the caspase-1 crystal structure used in our studies [34].

In this investigation, aerial parts of *Maytenus octogona* (ethanol extracts), employed in Peruvian ethnopharmacology as anti-inflammatory agents, were analyzed using a cyclovoltammetry method able to directly determine the scavenging of the superoxide radical. Leaf extracts were more active than those from stems. A comparison with other natural products analyzed by our group using the same protocol (olive oil [25], propolis [26] and black seed oil [27]) indicates slightly weaker scavenging for these extracts.

Recently, a study on the potential anti-inflammatory activity of some phenolic and methide-quinone *nor*-triterpenes isolated from *Maytenus retusa* that were evaluated for the inhibition of the NLRP3 inflammasome in macrophages appeared [35]. One of the most potent compounds evaluated was a semisynthetic methide-quinone Br derivative related to the three methide-quinones described in this work. This compound showed markedly reduced caspase-1 activity, IL-1β secretion (IC_50_ = 0.19 μM), and pyroptosis (IC_50_ = 0.13 μM). Since caspase-1 inhibition is important in preventing excessive cytokine production that can lead to inflammation, these findings can be associated with our results.

In this investigation, three natural methide-quinones derivatives contained in *Maytenus octogona* extracts were analyzed using molecular mechanics docking calculations. Results of these studies show the possible inhibition of caspase-1 by each of these compounds, thus supporting the anti-inflammatory medicinal use by indigenous populations in the related Peruvian area. Our work shows that Arg341 is the most important amino acid involved in inhibiting the active site of caspase-1. This amino acid is stabilized through strong intermolecular interactions for each of the three potential inhibitors, hydroxytingenone, tingenone, and pristimerin. The calculated binding energies of these three compounds are −13.7 kcal/mol for hydroxytingenone, −11.8 and 13.0 kcal/mol for tingenone, and −9.3 and −9.4 kcal/mol for pristimerin, which indicate that all three natural products are good inhibitors. Each of these methide-quinones is closely related to the methide-quinone Br derivative mentioned above [35]. That these interactions remain after the more rigorous molecular dynamics calculations underscores their significance. The methide-quinone carbonyl group in hydroxytingenone and pristimerin is preferred as a hydrogen bond acceptor over the other possible carbonyl acceptor located in these ligands, agreeing with results shown in a previous study [11]. This is confirmed by statistics on the 10 poses of pristimerin, showing only poses 1–3 having H-bond interactions with the O(carbonyl) located at the other end of the molecule, whereas for poses 4, 5, 7, 8, 10 the methide-quinone carbonyl is engaged, also in H-bonds. Tingenone shows both carbonyls having a similar preference due to closely related binding energy derived from poses 1 and 5. Our work supports the anti-inflammatory medicinal use of *Maytenus octogona* by native populations in Peru and provides impetus for studying the chemical mechanisms of additional established traditional remedies. Once again, we have shown that natural products can provide the chemical scaffold that targets important receptor sites such as caspase-1 in the inflammasome pathway for the development of a novel therapy against human diseases.

## Figures and Tables

**Figure 1 ijms-24-10750-f001:**
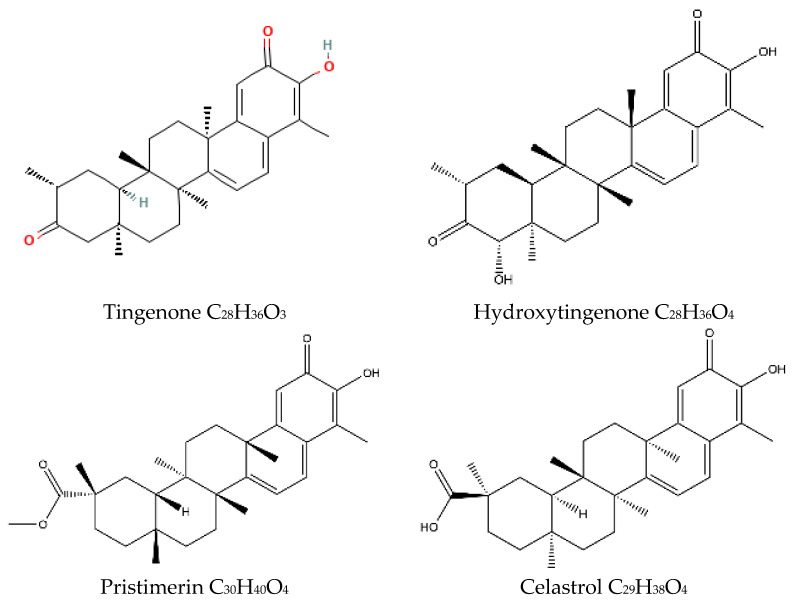
Two-dimensional canonical structural display of compounds analyzed in this study. Synonyms: hydroxytingenone = beta-hydroxytingenone; maytenin = tingenone.

**Figure 2 ijms-24-10750-f002:**
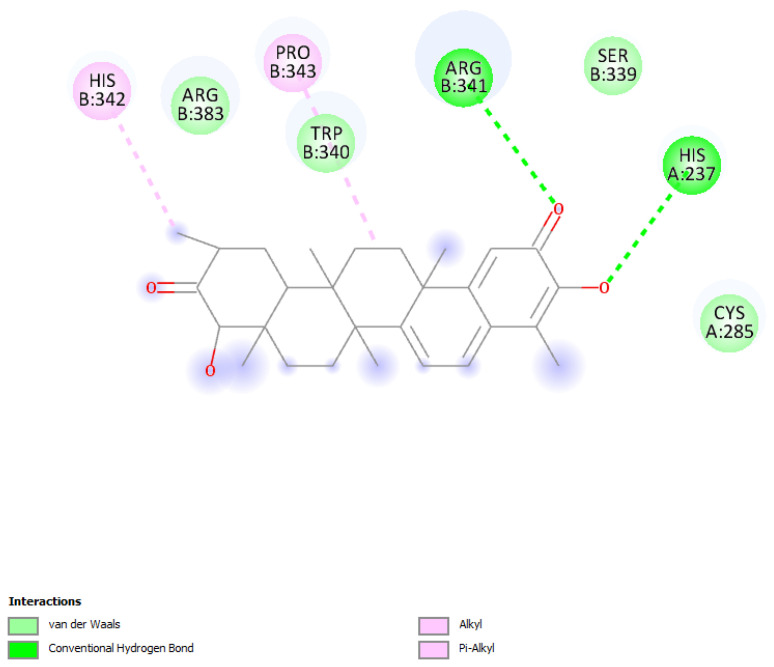
Pose 4 of hydroxytingenone docking, two-dimensional display. The methide-quinone carbonyl is engaged in hydrogen bond with Arg341 and His237, in contrast with the other carbonyl that shows no interactions.

**Figure 3 ijms-24-10750-f003:**
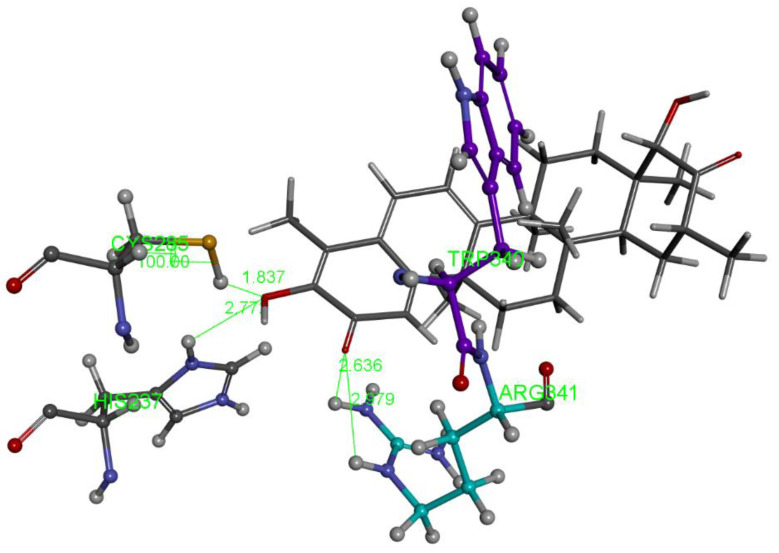
More detailed features of pose 4, hydroxytingenone, docking. The donor guanidino group of Arg341 (turquoise colored C atoms) has a double H-bond interaction with O(methide-quinone) carbonyl, 2.636 Å and 2.979 Å. The O(hydroxyl) associated with the methide-quinone inhibitor has a double H-bond interaction, one with H(Cys285), 1.837 Å, and one with H(His237), 2.772 Å (lower left of figure).

**Figure 4 ijms-24-10750-f004:**
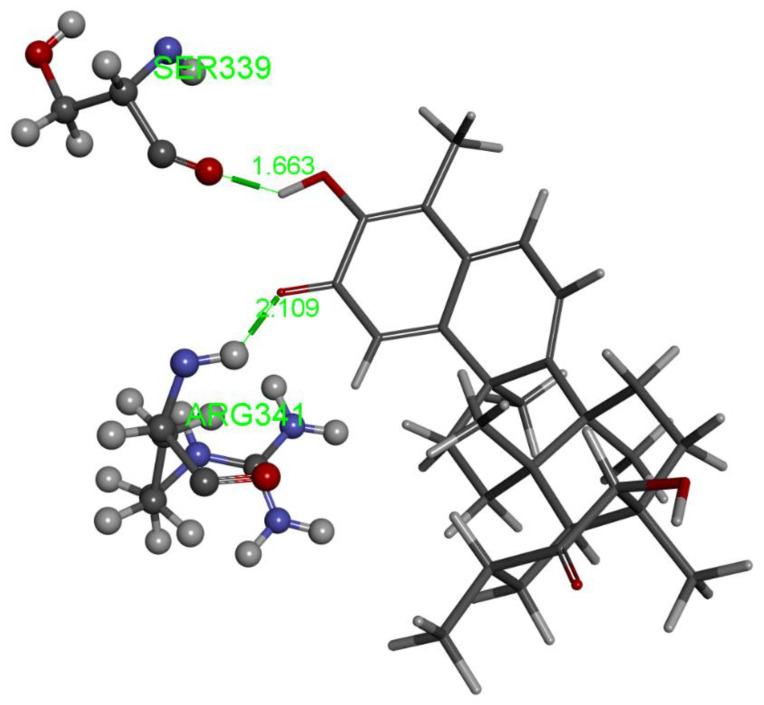
After molecular dynamics pose 4, hydroxytingenone has calculated binding energy of −13.7 kcal/mol. The H-bond interaction between the donor guanidino group of Arg341 and O(methide-quinone) carbonyl is confirmed and strengthened, 2.109 Å, whereas an H-bond of 1.663 Å by acceptor O(Ser339) replaces the former interactions by Cys285 and His237. A comparison with interactions shown by the inhibitor VX-765 at the crystal structure active site [22], shows involvement of both amino acids, Arg341 and O(Ser339), also forming hydrogen bonds; the latter interacts with a guanidino HN moiety. Indeed, the role of Arg341 appears very important as it has a double hydrogen bond interaction with two different O(carbonyl) moieties of VX-765.

**Figure 5 ijms-24-10750-f005:**
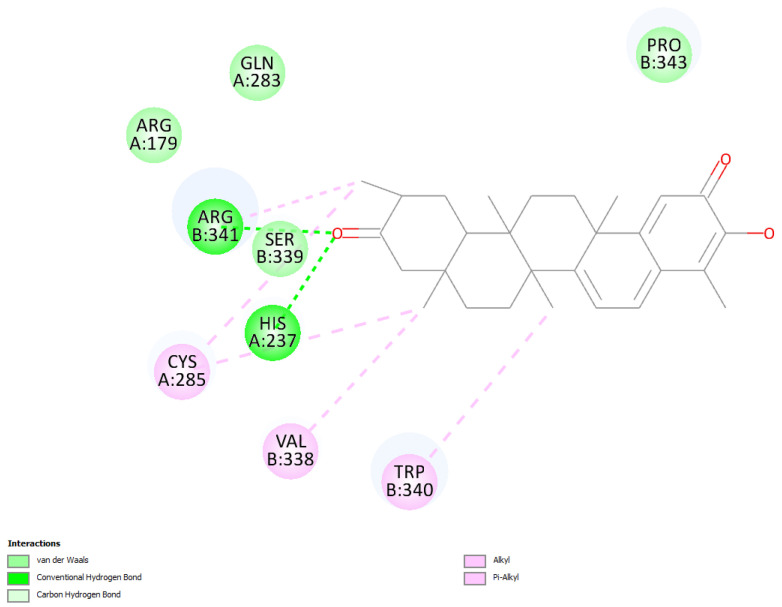
Pose 1. The tingenone Docking 2D, Arg341 forms a hydrogen bond to the non-methide-quinone carbonyl, which also has H-bond interaction with H(His237). The methide-quinone moiety appears free of interactions.

**Figure 6 ijms-24-10750-f006:**
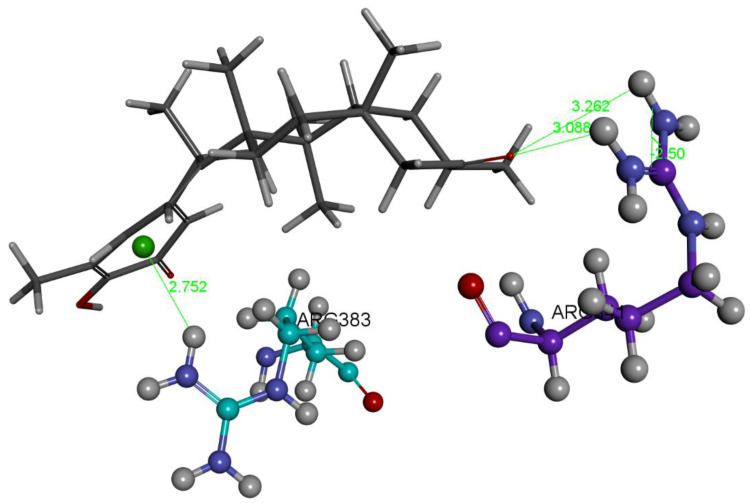
Pose 1. The tingenone dynamics of pose 1 has a calculated binding energy of −11.8 kcal/mol. The guanidino group of Arg383 (turquoise colored) establishes a cation-π interaction with the methide-quinone ring, 2.752 Å, whereas the originally docked Arg341 (violet colored) has a double non-chelating H-bond from the guanidino group to the opposite carbonyl, 3.262 Å and 3.088 Å. Interestingly, the cation-π interaction reminds us of the interaction that the same amino acid, Arg383, establishes with an aromatic ring of original inhibitor VX-765 in the crystal structure.

**Figure 7 ijms-24-10750-f007:**
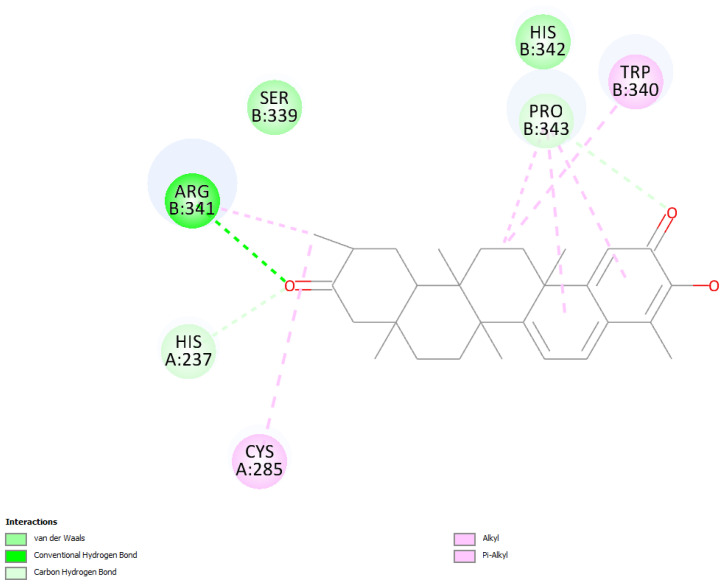
Pose 5. Tingenone docking 2D interactions. Arg341 establishes an H-bond to the non-methide-quinone moiety.

**Figure 8 ijms-24-10750-f008:**
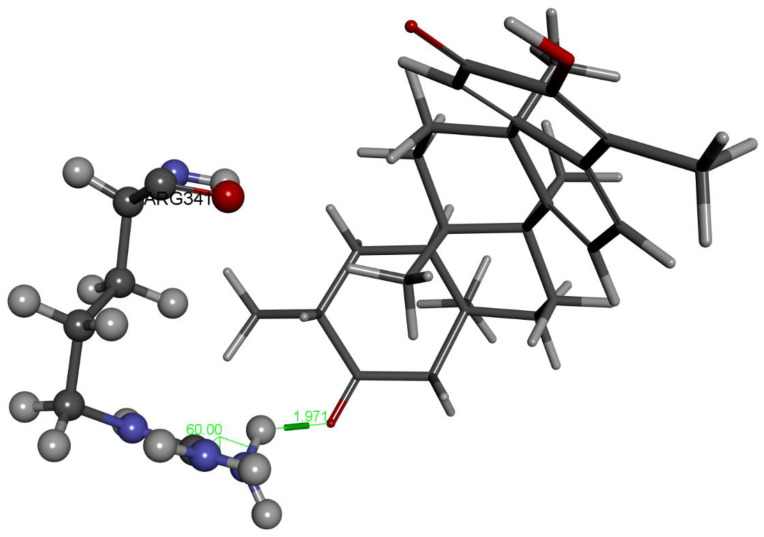
Pose 5. Tingenone dynamics has a calculated binding energy of −13.0 kcal/mol. It shows a very short H-bond distance of 1.971 Å, from guanidino H(Arg341) to the O(carbonyl) opposed to the methide-quinone carbonyl.

**Figure 9 ijms-24-10750-f009:**
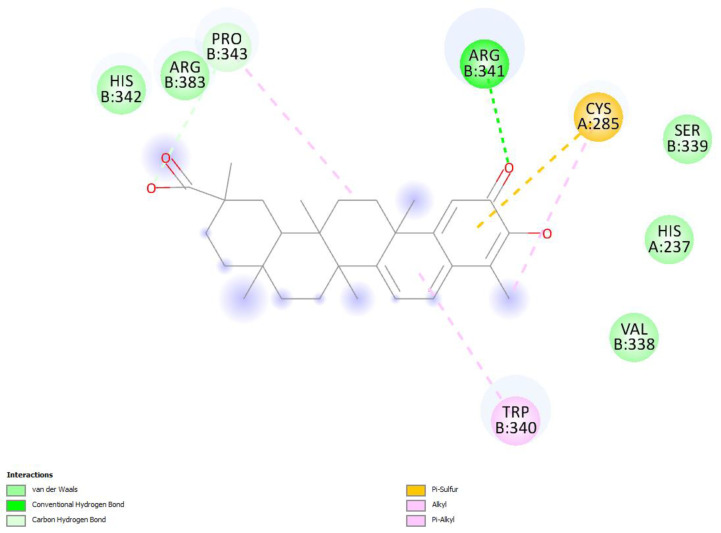
Pose 4. Pristimerin docking results, two-dimensional display. Cys285 shows π-sulfur interaction with the methide-quinone ring and Arg341 has a H-bond to the methide-quinone O(carbonyl).

**Figure 10 ijms-24-10750-f010:**
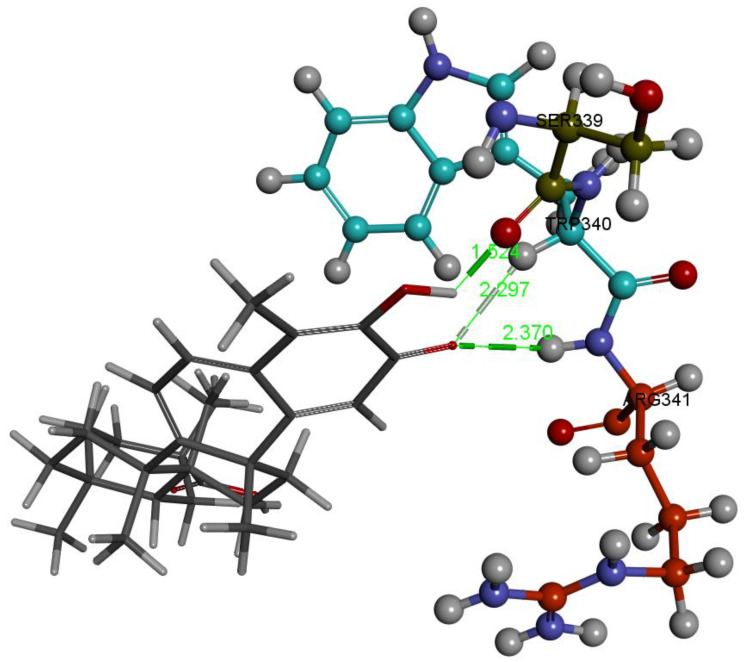
Pristimerin pose 4 dynamics after calculating the binding energy, −9.4 kcal. O(methide-quinone-carbonyl) establishes an H-bond to brown colored H(Arg341), 2.370 Å, and to turquoise colored H(Trp340), 2.297 Å, whereas O(Ser339) is 1.524 Å from pristimerin H(hydroxy). Interestingly, the above mentioned O(Ser339) interaction seen in hydroxytingenone, and found in VX-765 crystal structure through an HN moiety, is identical to that seen in pristimerin, supporting a potential role of this ligand and hydroxytingenone at the active site. In addition, the interaction between Trp340, indicated as π-alkyl in docking (Figure 9), and directed to the ring adjacent to the methide-quinone, transforms to hydrogen bonding with O(carbonyl) after dynamics, 2.297 Å.

**Figure 11 ijms-24-10750-f011:**
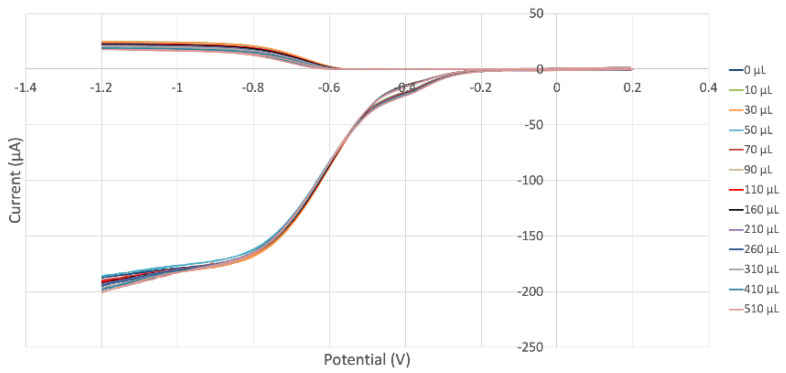
MORUnica (stem extract) RRDE voltammograms. Each run is associated with a specific color, for instance the 70 μL aliquot red line can be observed in the upper part, oxidation curve, detected at the ring electrode, which is consistent with the red line detected at the disk electrode for reduction during the same experiment and located at the bottom of the figure.

**Figure 12 ijms-24-10750-f012:**
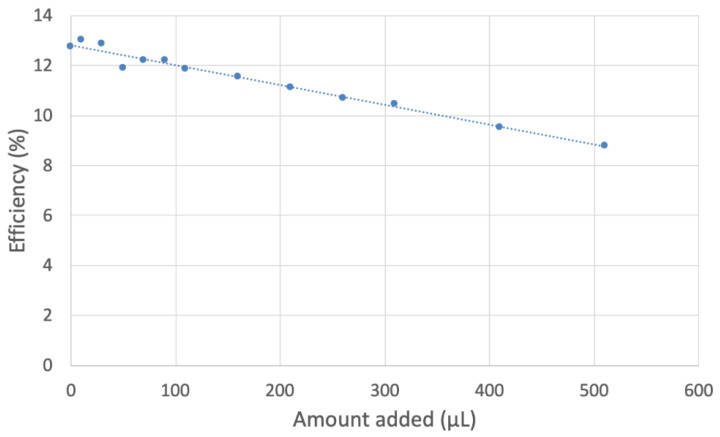
MORUnica collection efficiency shows a linear behavior, y = −0.008x + 12.807, R^2^ = 0.9757, whose slope is associated with the superoxide scavenging capability of the stem extract.

**Figure 13 ijms-24-10750-f013:**
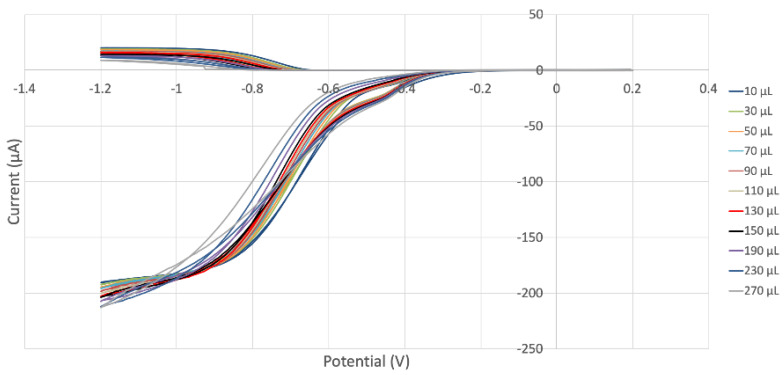
MOHUnica (leaf extract) RRDE voltammograms.

**Figure 14 ijms-24-10750-f014:**
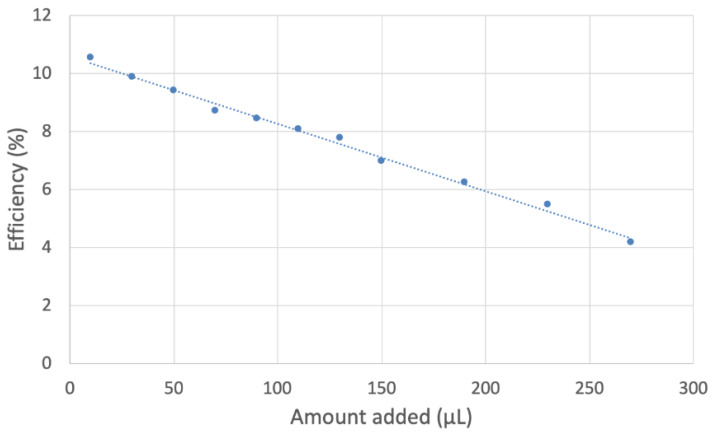
MOHUnica collection efficiency of the ethanol leaf extract, y = −0.0232x + 10.585, R^2^ = 0.994. Compared with the stem extract (Figure 12), the steeper slope in this figure indicates a stronger capability of scavenging superoxide.

**Table 1 ijms-24-10750-t001:** Amino acid H-bond and other interactions (*) with three components of *Maytena octogona* at the active site of Caspase-1 upon docking. “P” is for pose; “none” indicates only van der Waals interactions; “OUT” indicates that after dynamics the potential inhibitory molecule becomes displaced outside the active site. Numbers (kcal/mol) within each box represent binding energy after dynamics.

	P1	P2	P3	P4	P5	P6	P7	P8	P9	P10
H-Tingenone	Cys285His237 −7.2	Cys285 His237 −7.2	Arg341 Cys285 −8.9	Arg341 His237 −13.7	Trp340 π OUT	Trp340 π OUT	Trp340 OUT	Arg341 His237	Trp340 π OUT	Trp340 Cys285 (unfavored)
Tingenone	Arg341 His237 −11.8	His237 −11.8	His237 Cys285 −11.8	His237 −11.8	Arg341 * −13.0	His237 −11.8	Arg341 His342 (π-π) * −13.0	His237 Cys285 −11.8	Arg341 His342 (π-π) * −13.0	His237 Cys285
Pristimerin	Arg341 Cys285 OUT	Asp288 −9.3	Asp288 −9.3	Arg341 Cys285 (π-S) * −9.4	His342 (π-π) * −5.8	none −5.8	Arg341 Cys285 (π-S) * −9.4	His342 (π-π) * −5.8	none	Arg341 Cys285 (π-S) * −9.4

**Table 2 ijms-24-10750-t002:** Comparison of slopes of the plant products studied in this work with other natural products analyzed using the RRDE method.

Olive Oil [25]	Black Seed Oil [26]	Propolis [27]	MORUnica	MOHUnica
−0.0838	−0.078	−0.0864	−0.008	−0.023

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
