# Peer review of "Maytenus octogona* Superoxide Scavenging and Anti-Inflammatory Caspase-1 Inhibition Study Using Cyclic Voltammetry and Computational Docking Techniques"

_ijms, 2023, doi:10.3390/ijms241310750_

Round 1
Reviewer 1 Report (Previous Reviewer 1)
The revised version of this manuscript could be accepted for publication.
Author Response
Thanks for accepting the manuscript
Reviewer 2 Report (Previous Reviewer 2)
Comments to the Author
- It will be a good addition to the ethnomedicine field and will expand understanding of mechanism of action. Pertinent introduction, detailed experimental section and nice summary of results and discussion with supporting literature references provided. Previous feedback has been addressed.
- Minor title edit: Maytenus Octogona superoxide scavenging and anti-inflammatory Caspase-1 inhibition study using cyclic voltammetry and computational docking studies.
- Page 16 of 20, line 391: Table 1 should be Table 2, also without liquid chromatography–mass spectrometry (LC-MS) analysis and phenoldienone concentration tracking, a note should be added when comparing the various slopes of the different plant products for scavenging activity efficacy.
- Some figures (example Figure 3 and Figure 10) bond distances difficult to read, overlapping with image
- Doublecheck the amino acid interactions with various poses and bond distances in the entire document for accuracy
- Abstract: ‘It is worth noting that the methide-quinone carbonyl oxygen in these three compounds is the preferred hydrogen bond acceptor site’ Is this true for tingenone?
Author Response
See attached file

Reviewer 3 Report (New Reviewer)
The current version of manuscript is suitable for publication.
Author Response
Thanks for accepting the manuscript
This manuscript is a resubmission of an earlier submission. The following is a list of the peer review reports and author responses from that submission.
Round 1
Reviewer 1 Report
This manuscript is interesting for biochemistry community. The topic is definitely original and actual. The relationship between oxidative stress and inflammation is well known and exogenous antioxidants, primarily phytochemical natural products, may assist the body’s endogenous defense systems in preventing diseases due to excessive inflammation. In this study, authors evaluate the antioxidant properties of ethnomedicines from Peru that exhibit anti-inflammatory activity by measuring the superoxide scavenging activity of ethanol extracts of Maytenus Octogona aerial parts using cyclic voltammetry RRDE studies. The chemical composition of these extracts are known and the interactions of three methide-quinone compounds found in Maytenus Octogona with caspase-1 are analyzed using computational docking studies. Caspase-1 is a critical enzyme triggered during the activation of the inflammasome and its actions are associated with excessive release of cytokines. The most important amino acid involved in active site caspase-1 inhibition is Arg341 and, through docking calculations, authors see this amino acid is stabilized by the three potential methide-quinone Maytenus Octogona inhibitors, hydroxytingenone, tingenone and pristimerin. These findings also are confirmed after more rigorous molecular dynamics calculations. It is worth noting that the methide-quinone carbonyl oxygen in these three compounds is the preferred hydrogen bond acceptor site compared to the other carbonyl group present at the opposite side in these ligands. The results of these calculations and cyclovoltammetry studies support the effectiveness and use of anti-inflammatory ethnopharmacological ethanol extract of Maytenus octogona (L’Héritier) DC. The aim of the study is clear and the authors provided adequate information on how they conclude their results. The references are relevant and generally recent and include appropriate studies. It is clear what is already known about the topic. The research question is clearly outlined and justified. The process is valid and the variables are defined appropriately. The introduction provides sufficient background. The research methodology is adequate and modern. The results are clearly presented. The conclusions supported by the data. The manuscript good illustrated and interesting to read. English language and style are fine. I have only couple of minor suggestions:
- Topological analysis of the electron density distribution (QTAIM) could be useful tool for studies of noncovalent interactions in various chemical systems (from bio- to organic/inorganic/organometallic). It would be a good idea to cite in introduction some relevant papers about this or even utilize such technique in current or future projects: Nature Communications 2020 11 (1), 2921; Journal of Molecular Structure 2016 1111, 142-150; New Journal of Chemistry 2017 41 (9), 3246-3250; Inorganica Chimica Acta 2015 434, 31-36; Zeitschrift für Kristallographie-Crystalline Materials 2018 233 (6), 371-377; Inorganic Chemistry 2020 59 (23), 17320-17325.
- Some more detailed perspectives regarding the future research could be formulated in conclusions section.
Overall, this nice manuscript could be accepted for publication after minor revisions.
Reviewer 2 Report
Comments to the Author
1. It will be a good addition to the ethnomedicine field and will be very helpful for researchers in expanding their knowledge and mechanism of action. Pertinent introduction, detailed experimental section and nice summary of results and discussion with supporting literature references provided.
2. Title edit, too detailed: Maytenus Octogona plant extract superoxide scavenging and anti-inflammatory Caspase-1 inhibition study using cyclic voltammetry and computational modeling (or docking).
3. Give expansion of acronyms before using them (eg. RRDE in Abstract, PDB)
4. Check website Reference 27 is it in the correct format.
5. Even though it is mentioned ‘calculations did not include solvent effects' (Page 4 of 16, line 174), elaborate in discussion this and other limitations of the specific computational modelling.
6. Page 4 of 16, line 138: Whey these specific amounts were chosen? how is it related to the aliquot volumes in Fig 11 and 13, any phenoldienone concentration tracking? ‘Anhydrous DMSO (99.9% purity) solutions of MOH Unica 0.924 g and 0.103 g, 8.7 mlml of MOR Unica were the stock solution of analyzed samples.’
7. Similarly for Table 1 (Page 14 of 16) as well any phenoldienone concentration tracking for comparison of results?
Reviewer 3 Report
The study describes computer prediction of caspase-1 inhibition by three methide-quinone compounds from ethanol extracts of Maytenus Octogona. Regardless of the in silico work, the authors carried out cyclovoltammetric studies of an ethanol extract of the aerial parts of Maytenus octogona.
The article has a significant problem.
In the article, the authors did not show a direct relationship between docking calculations and RRDE voltammetry. If they did an in silico search for caspase-I inhibitors, they had to test it directly to see if Maytenus Octogona ethanol extracts themselves were capable of inhibiting caspase-1.
I think there are some kits available for such study.
I agree that the antioxidant capacity of plant extracts may explain their medicinal uses, but this has nothing to do with the first part of the manuscript.
Other problems:
Methods.
In the abstract, the authors wrote: These conclusions (results of the docking experiment) are also confirmed after a more rigorous molecular dynamics calculation. But there is no description of the MD experiment either in the methods or anywhere in the text.
Results and discussion.
3.1.1 and 3.1.2.
I think that for all substances, all 10 poses obtained in docking should be presented and discussed in some way in order to explain why the poses described in the text were chosen.
A final comparison of the activity of 3 quinone compounds should also be made and discussed.
Minor.
Fig. 2, 5, 7, 9. Color legend - not readable.
Fig. 3, 4, 6, 8, 10.
Headings of residues and distances should be moved to free places, and the color palette should be optimized.
Line 186. caspase-1 receptor?
Round 2
Reviewer 3 Report
Comments and Suggestions for Authors
The study describes computer prediction of caspase-1 inhibition by three methide-quinone compounds from ethanol extracts of Maytenus Octogona. Regardless of the in silico work, the authors carried out cyclovoltammetric studies of an ethanol extract of the aerial parts of Maytenus octogona.
The article has a significant problem.
In the article, the authors did not show a direct relationship between docking calculations and RRDE voltammetry. If they did an in silico search for caspase-I inhibitors, they had to test it directly to see if Maytenus Octogona ethanol extracts themselves were capable of inhibiting caspase-1.
Inflammation is characterized by a complex pattern of biological conditions and a pictorial view of these different factors has been shown by us in a previous paper, [Ref 11, Figure 1]. In this manuscript the subject is developed in the Introduction where it is mentioned that “A second signal, from various origins, involves releasing reactive oxygen species (ROS) that stimulate inflammasome assembly, leading to the activation of caspase-1 and the subsequent processing of pro-IL-1β into IL-1β and pyroptosis [3]”. Thus, once the enzyme is activated, a different action, enzyme inhibition, can avoid its role and decrease the cytokine storm. It is clear that the direct interaction between the active site of Caspase 1 and the potential pharmaceutical inhibitor, is not correlated with ROS, as ROS acts previously by contributing to Caspase-1 activation. Thus, to decrease the cytokine storm, it is positive to have an antioxidant that neutralizes the second signal due to ROS (avoiding caspase-1 activation), and it is positive to have an inhibitor of the activated enzyme. The proposed compounds have both functions (antioxidant to avoid ROS, shown by RRDE) and direct inhibition of Caspase-1 (shown by docking) and both are independent, but they contribute to decrease the cytokine storm, which is the subject of the special issue.
Regardless of previously published data, it is necessary to present a scheme of the various signaling pathways to abnormal inflammation that are considered in this work.
The current title “Maytenus Octogona superoxide scavenging and anti-inflammatory
Caspase-1 inhibition study using cyclic voltammetry and docking” need comma after inhibition
… they had to test it directly to see if Maytenus Octogona ethanol extracts themselves were capable of inhibiting caspase-1.
I think there are some kits available for such study.
Thank you for this valuable suggestion. We will try to do this in a future study
The validity of docking results that have not been experimentally verified is highly questionable, especially when they are poorly presented (see further comments).
I agree that the antioxidant capacity of plant extracts may explain their medicinal uses, but this has nothing to do with the first part of the manuscript.
As already mentioned, there are 2 independent actions involved and both are described in the Introduction.
It is necessary to present a scheme of the various signaling pathways to abnormal inflammation that are considered in this work.
Other problems:
Methods.
In the abstract, the authors wrote: These conclusions (results of the docking experiment) are also confirmed after a more rigorous molecular dynamics calculation. But there is no description of the MD experiment either in the methods or anywhere in the text.
A related description is now included in 2.4
Results and discussion.
3.1.1 and 3.1.2.
I think that for all substances, all 10 poses obtained in docking should be presented and discussed in some way in order to explain why the poses described in the text were chosen.
Cluster formation of poses is generally a good indicator of having found an effective inhibitor with higher probability. These features have been described in the manuscript when observed in the Docking outcome. On the other hand, some poses showed no important interaction with active site, as they have only van der Waals interactions, which are characterized by weak affinity, when compared with those described and these have not been included.
A comparative analysis of successful and unsuccessful poses can be given by a table that shows the gain in free energy and the number of contacts.
The difference between basic and MD-optimized posture is shown only for hydroxytingenone. Why?
A final comparison of the activity of 3 quinone compounds should also be made and discussed.
This is now included
Minor.
Fig. 2, 5, 7, 9. Color legend - not readable.
They are now enlarged conveniently.
Color legends are still not readable.
Fig. 3, 4, 6, 8, 10.
Headings of residues and distances should be moved to free places, and the color palette should be optimized.
Unfortunately, Discovery Studio graphics do not allow to relocate these items in more visible places, nor to modify distance color. To help understanding these items we clarified them in Figure captions. For instance, in Figure 6, Arg383 is turquoise colored, and relevant distances are mentioned.
1. I have no experience with Discovery Studio graphics. But the result of docking is not a picture but a structural model, i.e. a file that contains the three-dimensional coordinates of all atoms in the system, usually in text format. And I just can't believe that these files can't be edited to get clear pictures.
2. The captions to the figures, supposedly intended to clarify the situation, show that the authors do not have the necessary qualifications in the field. In order to describe the intramolecular and intermolecular interactions of amino acids in a protein, a uniform coding is used in structural biology, such as C-alpha, C-beta, ...You can't just say H, you have to tell which non-hydrogen atom it's bonded to.
At the moment, both the Figures and the descriptions are failed
Line 186. caspase-1 receptor?
This phrase has been changed in the manuscript to “caspase-1 in complex with a small molecule inhibitor”
Summary.
Unfortunately, during the revision, the authors failed to provide the necessary improvement (see comments). Therefore, I believe that the article cannot be published as it is.
Moderate editing of English language
Round 3
Reviewer 3 Report
I reiterate my proposal to reject the article because
1) Docking, the description of which is devoted to most of the article, which does not have any experimental confirmation in combination with the experimental part that is not directly related to docking, this is not the material that can be published in IJMS.
2) Figures representing docking results are suitable for internal use (lab seminar, student conference), but not for publication in IJMS.
3) The authors do not even try to respond constructively to criticism, answering like this:
never had any complaints before...
We can't improve the Figures....
Moderate editing of English language required